# Viral Components Trafficking with(in) Extracellular Vesicles

**DOI:** 10.3390/v15122333

**Published:** 2023-11-28

**Authors:** Félix Rey-Cadilhac, Florian Rachenne, Dorothée Missé, Julien Pompon

**Affiliations:** 1MIVEGEC, Université de Montpellier, IRD, CNRS, 34394 Montpellier, France; felix.rey-cadilhac@ird.fr (F.R.-C.); florian.rachenne@ird.fr (F.R.); dorothee.misse@ird.fr (D.M.); 2Faculty of Science, Université de Montpellier, 34095 Montpellier, France

**Keywords:** extracellular vesicles, viruses, infection, exosomes, microvesicles, cell-cell communication

## Abstract

The global public health burden exerted by viruses partially stems from viruses’ ability to subdue host cells into creating an environment that promotes their multiplication (i.e., pro-viral). It has been discovered that viruses alter cell physiology by transferring viral material through extracellular vesicles (EVs), which serve as vehicles for intercellular communication. Here, we aim to provide a conceptual framework of all possible EV-virus associations and their resulting functions in infection output. First, we describe the different viral materials potentially associated with EVs by reporting that EVs can harbor entire virions, viral proteins and viral nucleic acids. We also delineate the different mechanisms underlying the internalization of these viral components into EVs. Second, we describe the potential fate of EV-associated viral material cargo by detailing how EV can circulate and target a naive cell once secreted. Finally, we itemize the different pro-viral strategies resulting from EV associations as the Trojan horse strategy, an alternative mode of viral transmission, an expansion of viral cellular tropism, a pre-emptive alteration of host cell physiology and an immunity decoy. With this conceptual overview, we aim to stimulate research on EV-virus interactions.

## 1. Introduction to EVs and Viruses

For infectious agents, viruses are amongst the tiniest and simplest ones. They are made of a RNA or DNA genome packaged with capsid proteins and can be enveloped within an outer lipid bilayer anchored with envelop proteins [1]. Despite this constrained structure, their impact on human health is very concerning [2]. To name but a few, the infamous SARS-CoV-2 (virus abbreviations are detailed below the Tables) which caused the COVID-19 pandemic emerged in the human population at the end of 2019 and infected over 650 million people worldwide, resulting in 1% mortality rate, until the end of the WHO-triggered pandemic alert in 2022 [3]. The emerging mosquito-borne dengue virus threatens nearly half of the world population and infect approximately 400 million people yearly, causing more than 20,000 deaths [4,5]. Ebola and Nipah viruses are additional emerging threats with a very high fatality rate of 80 and 35%, respectively, albeit having lower transmission rates compared to the aforementioned viruses [6]. More alarmingly, the risks of viral emergence are amplifying due to human-driven global changes [7].

There are very few antiviral treatments, and medical support is usually limited to alleviating symptoms. Vaccination has been successful in several instances but can fail despite extensive efforts. For example, HIV-1 evades vaccine-induced immune responses [8] and dengue virus infection can be even enhanced by vaccine-induced immune responses [9,10]. In this context of increasing viral threats and lack of effective curative tools, a better understanding of virus-induced mechanisms that promote infection will reveal potential targets for novel interventions. A successful viral infection partially results from the ability of viruses to subdue host cells into creating an environment favoring virus multiplication, hereafter defined as pro-viral. Indeed, viruses need to evade antiviral immune responses [11] and alter the cellular machinery to complete their life-cycle [12,13]. Among the different interaction routes adopted by viruses to communicate with and influence host cells, extracellular vesicles (EVs) that enable cell-cell communication can mediate the transport of viral material [14].

EVs are non-replicative cell-derived membranous spherical structures secreted by most eukaryotic cells. They have an important role in intercellular communication by transferring their protein, lipid and/or nucleic acid cargo to other cells [15]. The size of EVs can range from 30 nm to more than 5 μm in diameter. EVs can be categorized in different populations based on size, content, and type of secreting cells. Nonetheless, the current consensus is to classify EV types according to their biogenesis [16]. It is first necessary to distinguish EVs produced by apoptotic cells that are called apoptotic bodies and range from 1 to 5 µm from EVs secreted by healthy cells. The latter cells secrete either ectosomes that range from 100 nm to several µm or exosomes that range from 30–150 nm [15,17]. Ectosomes [i.e., microvesicles (MV) and oncosomes] originate from budding of the plasma membrane. Exosomes are formed by inward budding of endosomal membrane, resulting in multivesicular bodies (MVB) which release their intraluminal vesicles in extracellular milieu upon fusion with the plasma membrane [17,18]. However, determining the biogenesis requires extensive efforts, and lack of this information sometimes leads to wrongly (as size is insufficient to categorize EV types) defining small EVs as exosomes [19]. Nonetheless, proteic and biophysical determinants can be used to approximate exosome identification. Presence of apoptosis-linked gene-2-interacting protein X (ALIX), Flotillin-1, tumor susceptibility 101 (TSG101) or tetraspanin proteins (e.g., CD63, CD9, CD81) in EVs ranging from 30 to 150 nm with a density of 1.1 to 1.19 g/mL on a sucrose gradient most likely indicate an exosome [20]. Of note, those criteria only apply to human exosomes, are not exhaustive and still debated. In the absence of extensive biophysical characterization, the latest classification guidelines distinguish small EVs (sEV) pelleted by centrifugation at ~100,000× *g* from large EV (lEV) pelleted at ~10,000× *g* [16]. Altogether, the generic term EV encompasses many different EV types that vary in their physical characteristics and contents.

Both EVs and viruses share similarities with regards to their secretion pathways, composition and function as inter-cell vehicles, although viruses are non-self, have a more complex structure, and can lack a lipidic envelope [21]. The physical similarities represent one of the biggest issues to study EVs in the context of viral infection as the biophysical similarities complexify isolation of EVs using gold standard density or size separation technologies (i.e., differential ultracentrifugation, density gradient, size-exclusion chromatography). For instance, one study claimed to detect structural proteins from dengue virions inside EVs that were isolated by ultracentrifugation from infected mosquito cell supernatant [22]. However, these results are questionable as dengue virion size and density are expected to overlap with those of EVs, resulting in EVs and virions being concurrently pelleted by ultracentrifugation [23]. Nevertheless, whenever antibodies targeting outer part of transmembrane proteins such as tetraspanins are available, EVs can be specifically purified by immunoprecipitation [24]. Alternatively, it is also possible to use immunoprecipitation to exclude the co-isolation of viral particles, although complete removal may be difficult to accomplish. Separation of EVs from viruses can be further complicated by the presence of EV proteins within the viral envelop [25] and alternatively viral structural proteins inside EVs [26]. Such hybrid compositions led to the hypothesis that infected cells can release a continuum of vesicles from pure infectious viral particles on one end to non-viral host EVs on the other end, with a range of EVs containing increasing amount of viral proteins in between these two extremes [21]. A better separation of EVs from viral particles can be achieved when EV and viral markers are well characterized as this is the case for HIV-1 in human infected cells [27]. Altogether, there is no universal methodology to separate EVs from viruses and successful separation is usually achieved by the use of antibodies recognizing well-characterized markers on the surface of either EVs or viruses [28].

In this review, we focus on EV-virus interactions and their impact on infection in the context of human viral infections (Figure 1). First, we recapitulate the different types of viral components that can be associated with EVs. Second, we describe the possible fate of EVs. Finally, we address the different effects that EVs loaded with viral material can have on infection. While we do not aim to be exhaustive with regards to EV-virus interactions throughout the different virus families, we instead wanted to provide an overview of the principles governing EV-virus crosstalk by collecting well-characterized evidence from different virus families.

## 2. Viral Components Contained in EVs

There are now multiple lines of evidence that EVs secreted from infected cells contain viral elements as cargo [29,30]. Information about how viruses modify EV biogenesis is not provided here and can be found in a previous review [14]. Below we detail the different types of viral components that were found with(in) EVs (Table 1).

### 2.1. Entire Viral Particles

The most complex viral elements that can be found in EVs are single or multiple particles. This association was mainly reported for naked viruses of the *Picornaviridae* family. While secretion of these naked viruses can be achieved through host cell lysis, viral particles from HAV and CVB3 were detected in ectosomes, which are produced through plasma membrane budding [31,32]. Clusters of enteroviral PV particles were also detected in phosphatidylserine-enriched vesicles [33] and JCPyV particle were found inside lEVs and sEVs [34,35]. Nevertheless, secretion of multiple or single viral particles into EVs is not limited to *Picornaviridae* and naked viruses. It was estimated that a dozen of BKPyV or multiple enveloped SARS-CoV-2 particles are secreted inside single EVs [36,37]. Similar observations were made for the enveloped HIV-1 and HCV viral particles [38,39,40]. 

Mechanistically, virions can be internalized by EVs during EV biogenesis via direct interactions of viral surface proteins and proteins involved in EV biogenesis (we refer to a previously-published extensive review on EV biogenesis to describe such proteins [15]). For instance, HAV capsid proteins interact with ALIX to hijack host endosomal sorting complexes required for transport (ESCRT) [31,67]. As a result, the non-enveloped HAV can also be secreted inside an EV-like membrane. Alternatively, viruses can subdue autophagosomes for secretion. For instance, enteroviruses internalized by autophagosomes can avert lysosomal degradation by promoting the fusion of autophagosomes to plasma membranes. This results in the formation and release of phosphatidylserine-rich unilamellar vesicles containing these viruses [33]. 

### 2.2. Viral Proteins

During cell infection, structural and non-structural viral proteins can be directed to EVs. Enveloped viruses contain structural glycoproteins within the envelope compartment. These glycoproteins have functions in receptor recognition and cell entry, and were found in EVs released from cells infected by several viruses. The Spike protein of SARS-CoV-2 [41], and the envelope protein of HIV-1 [26] and ZIKV [24] were detected in uncharacterized EVs, while the Ebola virus glycoprotein was found in the ectosomal fraction of EVs [43]. The presence of these surface antigens in both EVs and viruses illustrates the difficulty to isolate EVs by immunocapture. Other structural proteins such as capsid proteins that directly interact with viral genomes can likewise be found in EVs. For example, the structural Gag protein of HIV-1 [42] or Ebola’s VP40 structural protein are secreted in EVs [44].

Aside from structural proteins, viruses encode non-structural proteins with functions in viral cellular cycle that are also transported by EVs. The best documented case is the incorporation of the HIV-1 Nef accessory protein into sEVs that exhibit markers and the density of exosomes [45,46,47]. Other interesting findings include the incorporation of the EBV LMP-1 protein into sEVs [48,49]. This protein modulates cellular gene expression and contributes to human cancer development. The HBV HBx protein that perturbs the host cell cycle, potentially causing hepatocellular carcinoma, was also found in sEVs likely corresponding to exosomes [50]. Similarly, HPV-16 E6 and E7 protein were associated to EVs [52]. It is possible that the ZIKV Non-Structural 1 (NS1) protein associates with EVs by binding to their surface [51]. These different examples illustrate the broad diversity of viral proteins secreted in associations with EVs and provide clues on the functional variability of EVs that contain viral proteins. 

Mechanistically, as for virion internalization, EV loading can result from a direct interactions between viral proteins and cellular proteins involved in EV biogenesis. For instance, the EBV LMP-1 protein binds to CD63 tetraspanin and enables LMP-1 EV loading [68]. Additionally, it was observed that higher-order oligomerization of proteins increases sorting of the resulting protein aggregates into MVB [69], although the mechanism is unclear. Informatic modelling also identified protein sequences and post-translational modifications that may promote EV loading [70,71,72]. However, there is no experimental evidence for such a scenario. Eventually, the wide variability of protein content in EVs prompted some authors to propose that EV loading is a stochastic phenomenon [73].

### 2.3. Viral Nucleic Acids

Different types of viral nucleic acids with different functions are produced during infection and found in EVs. The incorporation of whole RNA or DNA genomes into EVs is described for many viruses, although most of the available literature concerns RNA viruses. The positive-sense genomic RNA (gRNA) of SARS-CoV-2 [53], HIV-1 [54] and HGV [56], and both the positive-sense and replication-competent negative-sense gRNA of HCV [55,74] were detected inside EVs. Moreover, entire DNA-based viral genome are secreted into sEVs, as observed for HBV [57,58] and oncolytic Adenoviruses [59]. Parts of the HPV-18 DNA genome also seem to be associated with sEVs [60].

Viral mRNA fragments produced by transcription of DNA genomes or by splicing of RNA genomes have been identified in EVs and can serve as templates for protein translation once internalized. With regard to DNA genome viruses, mRNAs coding for the HBx protein of HBV [50] or for the LMP-1 protein of EBV [61] were found in sEVs. Concerning RNA viruses, RNA coding for the Gag p17 protein of HIV-1 [54] or for multiple proteins of HSV-1 [62] are secreted into sEVs. 

EVs can harbour non-coding viral RNAs associated with various functions. MicroRNAs (miRNAs) serve as post-transcriptional regulators of gene expression and are produced by viruses to alter cellular or viral processes [75]. For example, HIV-1 secretes VmiR88 and VmiR99 into EVs to activate immunity [63] and the transactivating response element (TAR) miRNA into sEVs to inhibit apoptosis in recipient cells [64]. Into sEVs, HSV-1 secretes miR-H3, H5 and H6 that have potential functions in curtailing viral spreading [62], while EBV secretes multiple miRNAs that target cellular mRNAs [65,76]. Moreover, EVs contain longer non-coding viral fragments. Our group recently reported that a subgenomic flaviviral RNA (sfRNA), resulting from partial degradation of the flavivirus RNA genome, may be secreted via mosquito salivary EVs [66]. This sfRNA has immune inhibitory properties [77] and when secreted via saliva was associated with enhanced infection and hence viral transmission.

Mechanistically, viral nucleic acid sequences can be packaged into EVs by direct interaction with EV-biogenesis proteins or indirect interactions with RNA-binding proteins (RBPs) that associate with EV biogenesis proteins. In support of the first scenario, some proteins of the ESCRT are endowed with RNA-binding properties [78] and interaction of viral RNA with ESCRT proteins has been reported for the non-enveloped tomato bushy stunt virus (TBSV) [79]. In support of the second modality, it has been observed that EVs are enriched in RBPs, including dsRNA-binding proteins [80]. DsRNA-binding proteins can interact with structured viral RNA [81] or bind to dsRNA resulting from miRNA binding to homologous viral RNA. For example, HCV gRNA is bound by miR-122 to attract the dsRNA-binding protein Ago2 [55]. The resulting dsRNA-Ago2 complex is then stabilized by HSP90, which regulates EV biogenesis and loading [82]. Interestingly, specific nucleic acid sequences are also suspected to promote nucleic acids internalization into EVs, and a 12-nucleotide RNA sequence has been informatically identified as a determinant of EV sorting [83], although experimental evidence is lacking.

## 3. Fate of Viral Components Associated with EVs

After their release into the extracellular space, EVs can either target neighbouring cells or circulate in biofluids to reach more distant recipient cells.

### 3.1. Biofluid Dissemination

Multiple studies observed the circulation of ectosomes and exosomes in human biofluids. Initially found in blood [84,85], EVs were later detected in human saliva, breast milk [86], urine [87], semen [88], synovial fluid [89] and cerebrospinal fluid [90], indicating the ubiquity of EV circulation. The presence of EVs in biofluids can enable an inter-organ transmission of their cargo, which can contain viral material. Such long-range transport was observed for EV-associated PD-L1 protein secreted by cancer cells in plasma and resulted in systemic immunosuppression and enhancement of tumor growth in distant organs [91]. Nonetheless, the biodistribution of EVs is not homogeneous among the different organs. After intravenous inoculation of mice, EVs are mostly internalized by the liver, spleen, lungs and kidneys, although EVs are also found in brain, heart, bone or bladder [92]. The variability of the biodistribution of EVs is determined by multiple parameters such as the cell origin and the EV type. Bodyfluids can be easily collected by non- or minimally invasive procedures, allowing expedient isolation of EVs. Studies on the use of biofluid-derived EVs as potential diagnostic markers are under progress [93], notably in oncology [94,95]. In addition, this approach was recently proposed for COVID-19 diagnostics by the detection of EVs carrying SARS-Cov-2 Spike protein [96]. 

### 3.2. Cellular Fate

EVs are able to interact with recipient cells to deliver their cargo through several pathways, although the detailed molecular mechanisms remain elusive. Internalization is initiated by interactions between EV surface proteins and cellular receptors that activate an intracellular signaling cascade, resulting in EV uptake. EVs can either directly fuse with cellular plasma membranes to release their content into the cytoplasm or enter the cells by macropinocytosis, phagocytosis or endocytosis mediated by clathrin, caveolin or facilitated by lipid rafts [15,97]. Within endosomes, EVs can fuse with the endosome membrane to release their content into the cytoplasm [98]. Alternatively, EVs can be directed to lysosomes for degradation. In the absence of viral infection, it was observed that miRNA and mRNA were transferred through plasmatic EVs, leading to the translation of incoming mRNA in recipient cells [99,100].

It is widely accepted today that EVs are internalized by recipient cells. However, the efficacy of biomolecule delivery by EVs has been recently challenged using improved isolation techniques and more refined analytical tools: Bonsergent et al. observed that only 1% of EVs harbouring tagged cytosolic cargo are uptaken within the first hour of exposition by mammalian cells [101], although the internalization rate is most probably dependent on the cell and EV types. Furthermore, only 35% of the internalized EVs released their cargo into the cytosol, suggesting an overall low efficiency of EV-mediated delivery, at least under in vitro conditions. Interestingly, in the same study, the authors suggested that EV internalization occurs through endocytosis, is probably mediated by a fusion peptide and that EV release in endosomes is dependent on acidification. All these characteristics are shared by some viruses. It is also tempting to speculate that the presence of viral fusion proteins in virus-host hybrid EVs can enhance EV delivery, as shown for engineered EVs that contained vesicular stomatitis virus-derived membrane fusion protein [102].

Alternatively, EV’s fate is not necessarily related to cell cargo delivery. For instance, EVs may interact with circulating antibodies as described for EVs secreted from Zika virus-infected cells that display immunogenic viral envelope proteins [24]. 

## 4. Functions of EVs Containing Viral Material

How EVs with viral components influence infection depends on the virus species, the EV-transported viral element, the type of EVs and the recipient cell. In an effort to summarize this multifactorial scenario, we distinguish five classes of EV-dependent pro-viral strategies (Table 2). 

### 4.1. The Trojan Horse Strategy

Viral particles are hidden by a host cell-derived lipid bilayer within EVs. This metaphorically reminds of the Greek soldiers lurking inside a wooden horse to penetrate the impregnable city of Troy [20,109]. The EV lipid bilayer prevents detection of viral surface proteins by the immune system, facilitating viral infection and spread [31,34,37,39]. Additionally, EVs can include multiple viral particles, thereby increasing the multiplicity of infection upon delivery in cytoplasm [33,36,103]. A higher virus load can overwhelm immune defense and result in increased transmission and disease severity. For non-enveloped viruses, EV secretion represents a non-lytic egress mechanism that possesses the same pro-viral advantages as described above [33]. 

### 4.2. An Alternative Mode of Viral Transmission

Viral transmission can be achieved by EV-mediated transfer of viral genomes without any other viral material [59,110]. For instance, EV-mediated delivery of RNA virus genomes was sufficient to induce a productive infection in recipient cells for DENV, HCV and pegivirus [22,56,74]. Either positive or negative strand gRNA can be released and initiate viral protein translation or replication, respectively, as shown for HCV [55,111,112]. In addition to viral genomes, EVs can also contain viral cofactors (i.e., proteins, non-coding RNA species) that enhance EV-mediated viral propagation [113]. Nevertheless, experimental detection of viral genomes inside EVs has to be cautiously controlled to ascertain effective elimination of virions during EV purification. As detailed for the Trojan horse strategy, EV-mediated virion-independent mode of transmission may also increase the multiplicity of infection when occurring simultaneously to virion infection and reduce immune recognition.

### 4.3. Expanding Viral Cellular Tropism

Viral cellular tropism is determined by virus ability to enter specific cells and may vary when viruses are associated with EVs. Virus cell entry depends on the presence of specific receptors on the cell surface that interact with viral ligands and/or by specific proteolytic activation of viral ligands [114,115]. Therefore, each virus species can only enter a limited number of cell types that express the required binding partners. However, the cellular tropism of viruses found inside EVs is no longer dependent on viral surface proteins but determined by EV properties. For instance, human Polyomavirus JCPyV virions packaged in EVs are able to infect brain cells even though the cell type lacks the required surface receptors for viral attachment and entry [35]. Moreover, EV-mediated transfer facilitates virus diffusion via biofluids to reach distant and sometimes immune-privileged organs. Although not demonstrated in the context of infection, EVs could enable brain infection by providing a vehicle to translocate through the blood-brain barrier [104]. 

### 4.4. Pre-Emptive Alteration of the Cell Physiology

EVs can deliver different types of viral components to neighbouring cells or distant tissues to reconfigure cellular environments, most of the time in favor of viral infection. Various cellular mechanisms can restrict infection and EVs secreted by already infected cells can alter these cellular control mechanisms to facilitate viral dissemination to neighboring cells. One of the most telling and documented example concerns the HIV-1 accessory protein Nef that is found inside EVs [116]. This EV-viral protein association promotes viral dissemination in multiple ways: (i) delivery of Nef to CD4+ T cells provokes apoptosis and immune cell population decay resulting in hampered immune surveillance [46,105]; (ii) activation of resting CD4+ T cells, which are metabolically resistant to HIV-1 infection, renders them permissive to infection [47]; and (iii) alteration of B cells enables virus evasion from humoral immunity [106]. Furthermore, expression of Nef induces EV biogenesis and secretion, thereby acting as a positive loop signal to amplify Nef’s own secretion [107]. Aside from favoring viral dissemination by altering immune responses, EVs containing viral proteins can influence pathogenicity [43,44,117]. HBV HBx and EBV LMP-1 proteins secreted in EVs increase HBV-associated liver diseases [50] and EBV-induced carcinomas (e.g., Burkitt’s lymphoma and nasopharyngeal carcinoma) [49,108]. Messenger RNA coding for pro-viral proteins can also be delivered by EVs and, following translation, result in increased dissemination or pathogenicity, as observed for mRNAs coding for HBx or LMP-1 [50,61]. 

EVs can also transfer viral non-coding RNAs that regulate gene expression or interact with specific proteins that affect neighbouring cells and/or other cell types. EVs carrying HIV-1 miRNA and TAR element RNA activate CD4+ T cells to promote viral replication in recipient cells [63,64]. EV-mediated delivery of HSV-1 miRNA, however, diminishes viral replication and dissemination, although this may benefit the virus in the longer term by inducing virus latency to temporarily evade immunity [62]. Moreover, longer viral non-coding RNAs can be transferred by EVs to enhance infection by inhibiting immune responses. For instance, mosquito-borne DENV produces a non-coding RNA (i.e., sfRNA) with immune inhibitory properties that seems to be delivered into skin cells by mosquito salivary EVs during biting [66]. SfRNA interactions with proteins involved in immune responses [118,119] could then promote skin infection and increase viral transmission.

### 4.5. A Decoy for Humoral Immunity

As part of humoral immunity, antibodies or complement molecules can bind surface proteins of the viral envelope or capsid to prevent protein-protein interactions with cell receptors and inhibit cell entry. EVs that display such immunogenic viral proteins on their surface can react with the corresponding antibodies to sequester them, thereby alleviating the immune pressure on viruses [41,43]. Nonetheless, antibody decoy can also reduce infection intensity when sub-neutralizing antibodies improve infection through antibody-dependent enhancement (ADE) [24]. 

## 5. Discussion

In this review, we aim to summarize how EV-virus interactions can shape infection. In particular, we highlight the wide range of viral components that can be associated with EVs, demonstrating that virtually all types of viral components can be detected in EVs. We present how EV packaging influences the fate of viral components by documenting that EV packaging augments both the range of and distance to cell targets. Finally, we outline the various pro-viral strategies resulting from EV packaging, illustrating a toolbox accessible to viruses to promote infection. 

Our review is probably incomplete as EV biology remains a relatively recent field and improvement of EV purification technologies and knowledge will surely reveal new insights into EV-virus interactions. In this context, we would like to emphasize the importance of isolating characterized pure EVs, specifically devoid of viruses, to study EV functions on viral infection. Biophysical similarities between virus particles and EVs result in mixed populations when using most of the currently-available isolation technologies. Furthermore, it is difficult to characterize the large diversity of EV subtypes with the limited detection thresholds of current technologies. Nonetheless, characterization of multiple subtypes was achieved using multidimensional correlative proteomics in the context of HIV-1 virus infection in human cells [27]. The same study also unveiled the existence of hybrid populations of virions and EVs, containing variable proportions of host and viral proteins, further complicating isolation of pure particle populations with strictly either viral or host EV proteins. Eventually, a large majority of papers on EVs studies human EVs, drastically limiting our understanding of EV-virus interactions in biology. Recent works in arthropods such as mosquitoes and ticks show that viruses in other phylum also harness EVs to promote infections [66,120]. EV knowledge from other models will surely enrich the field of EV biology. 

Infection-triggered changes in EVs which do not contain viral material may also influence infection as EVs have roles in many diverse biological functions relevant to viral infections. Among others, the immune system is regulated by EV-mediated intercellular communication and can reduce viral infections [29,121,122]. The transfer of metabolic components through EVs alters cellular metabolisms [123], which is critical for viruses to complete their life-cycle. EVs can also regulate cell death [124] and thereby alter infections. Examples of viruses interfering with EV biogenesis include HIV-1 Gag accessory protein that recruits TSG101 [125], which is essential for late endosomal budding during exosome biogenesis, and Ebola virus VP40 protein that regulates exosome biogenesis by upregulating ESCRT-II, CD63 and Alix protein expression [44]. Given the reported interactions between viruses and EV biogenesis [14], viruses most probably also influence infection output by regulating secretion of EVs without viral material but with functions in biological processes relevant to viral infections. 

The capacity of EVs to transfer material between cells questions biology and virology dogmas. In biology, cell types are partially defined by their contents. EV-mediated trafficking of proteins, lipids and nucleic acids between cells alters cellular content, therefore blurring the distinction between different cell types. In the context of viral infections, permissive cells are distinguished from non-permissive cells by the presence of specific receptors for viral ligands or by their metabolic and immunologic susceptibility to infection. EV-mediated fluidity of cellular receptors and immune regulators can shift non-permissive cells to a permissive status [126,127]. Moreover, in virology, there is a physical distinction between enveloped vs. non-enveloped viruses. However, the capacity of certain viruses to hijack EV biogenesis to acquire a lipid-based envelope [128] complicates the physical segregation between enveloped and non-enveloped viruses. Altogether, the biophysical similarities and shared biogenesis pathways between EVs and viruses suggest an evolutionary link, which could provide a rationale for the many examples of EV-virus associations.

## Figures and Tables

**Figure 1 viruses-15-02333-f001:**
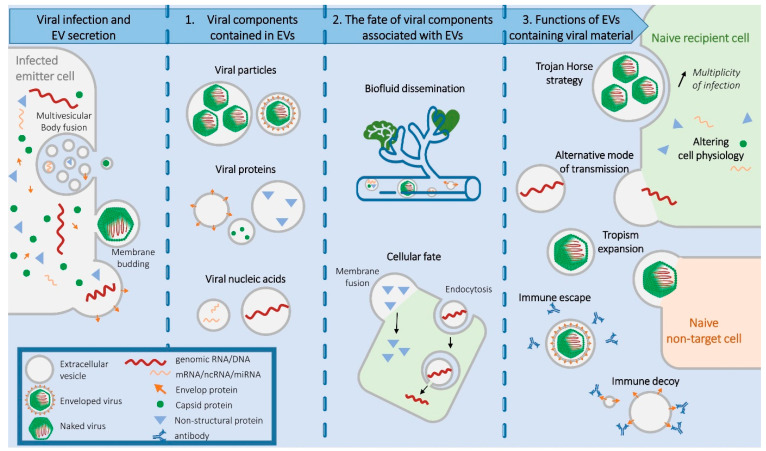
Overview of the associations between Extracellular Vesicles (EVs) and viral components. Each column separated by a dotted line illustrates a section of the text. The first column shows the different viral components present in an infected cell and the main EV secretory pathways, as detailed in the introduction. The second column indicates the types of EV-contained viral elements, as presented in Section 2. The third column outlines EV trafficking and delivery of viral cargo to naïve cells, as explained in Section 3. The fourth column depicts the different categories of functions of EVs containing viral material, as detailed in Section 4. Naive recipient cells represent non-infected cell permissive to canonical viral entry. Naive non-target cells represent cells that are not permissive for viruses. mRNA, messenger RNA; ncRNA, non-coding RNA; miRNA, micro-RNA.

**Table 1 viruses-15-02333-t001:** List of viral components within extracellular vesicles.

Viral Cargo	Type of Cargo	Examples	Information about the Type of EV (Methods of Purification and Analysis)	Reference
Virions	Naked viral particles	HAV	Uncharacterized EV type but distinct from viral particle (separation by iodixanol density gradient)	[31]
		CVB3	Uncharacterized EV type but distinct from viral particle (separation by iodixanol density gradient and visualization by EM)	[32]
		PV	Large phosphatidylserine-enriched vesicle (pelleted at 5000 g pellet and visualization by EM)	[33]
		JCPyV	sEV (Pelleted at 100,000 g and visualization with EM)	[34,35]
	Enveloped viral particles	BKPyV	Uncharacterized EV type but distinct from viral particle (separation by iodixanol density gradients and visualization by EM)	[36]
		SARS-CoV-2	lEV (pelleted at 400× *g* and visualization by EM)	[37]
		HIV-1	Putative exosome and ectosome (isolation by immunoaffinity and sucrose cushion centrifugation)	[38]
		HCV	sEV related to exosomes (pelleted at 64,000 g with sucrose cushion and characterization by marker detection)	[39,40]
Proteins	Structural proteins	SARS-CoV-2 Spike protein	Uncharacterized EV (Isolation by tangential flow filtration and EM/Nanoscale flow cytometry)	[41]
		HIV-1 Envelope protein	Uncharacterized EV type but distinct from viral particle (Isolation by magnetic nanoparticles capture)	[26]
		HIV-1 Gag protein	sEV related to exosomes (Pelleted at 70,000 g and isolated by sucrose gradient and identification by exosome markers)	[42]
		ZIKV Envelope protein	Uncharacterized EV type but distinct from viral particle (Isolation by sequential ultracentrifugation and affinity capture)	[24]
		EBOV GP protein	Uncharacterized EV (Isolation by 21,000× *g* centrifugation and sucrose gradient)	[43]
		EBOV VP40 protein	Uncharacterized EVs (Isolation by nanotrap particle capture)	[44]
	Non-structural proteins	HIV-1 Nef accessory protein	sEV related to exosomes (Pelleted at 70,000 g to 400,000 g in different studies and association with exosome markers)	[45,46,47]
		EBV LMP-1 protein	sEV (Pelleted at 70,000 g)	[48,49]
		HBV HBx protein	sEV (Pelleted at 110,000× *g* with sucrose cushion centrifugation)	[50]
		ZIKV NS1 protein	Uncharacterized EV related to exosome (Isolation with exosome isolation kit, Size exclusion chromatography, PEG purification and ELISA capture)	[51]
		HPV-16 E6 and E7 protein	Uncharacterized EV related to exosome (Isolation with Size exclusion chromatography, EM visualization and marker detection)	[52]
Nucleic acids	Genomes	SARS-CoV-2 gRNA	Uncharacterized EV (Isolation by Total Exosome Isolation Reagent)	[53]
		HIV-1 gRNA	sEV (Pelleted at 70,000 g with iodixanol density gradient)	[54]
		HCV gRNA	Uncharacterized CD63+ EV (Isolation with exosome isolation kit and capture with CD63 immuno-magnetic beads)	[55]
		HGgV gRNA	Uncharacterized EV (Isolation with Total Exosome Isolation Reagent)	[56]
		HBV gDNA	sEV (Pelleted at 110,000 g and visualization by stimulated emission depletion microscopy)	[57,58]
		oncolytic AdV gDNA	Uncharacterized EV type but distinct from viral particle (Isolation by iodixanol density gradients and visualization by EM)	[59]
		HPV-18 parts of gDNA	sEV (Pelleted at 95,000 g and visualization by EM)	[60]
	Messenger RNAs	HBV HBx mRNA	sEV (Pelleted at 110,000× *g* with sucrose cushion centrifugation)	[50]
		EBV latency-associated protein mRNA	sEV (Pelleted at 100,000 g and isolation with CD63 immunoprecipitation)	[61]
		HIV-1 Gag P17 mRNA	sEV (Pelleted at 70,000 g with iodixanol density gradient)	[54]
		HSV-1 ICP27, VP16 and LAT mRNA	sEV different from viral particle (Pelleted at 70,000 g with Dextran 10 density gradient)	[62]
	Non-coding RNAs	HIV-1 vmiR88, vmiR99 and vmiR-TAR	Uncharacterized EV (Isolation with exosome isolation kit and CD63 detection)	[63]
		HIV-1 TAR RNA	sEV related to exosome (Pelleted at 100,000 g with iodixanol density gradient and characterization with marker detection)	[64]
		HSV-1 miR-H3, miR-H5 and miR-H6	sEV different from viral particle (Pelleted at 70,000 g with Dextran 10 density gradient)	[62]
		EBV EBER-1 and EBER-2 RNA	CD63+ sEV (Pelleted at 100,000 g, characterization with marker detection and visualization with EM)	[65]
		DENV2 sfRNA	Uncharacterized EV different from viral particle (RNase protection essay, Visualization with super-resolution microscopy combined with RNAish)	[66]

Virus abbreviations: AdV, Adenovirus; BKPyV, BK Polyomavirus; CVB3, Coxsackievirus B3; DENV2, dengue 2 virus; EBOV, Ebola virus; EBV, Epstein-Barr virus; HAV, Hepatitis A virus; HBV, Hepatitis B virus; HCV, Hepatitis C virus; HGV, Hepatitis G virus; HIV-1, Human Immunodeficiency Virus 1; HPV-16, Human Papillomavirus 16; HPV-18, Human Papillomavirus 18; HSV-1, Herpes simplex 1 virus; JCPyV, JC polyomavirus; PV, Poliovirus; SARS-CoV-2, severe acute respiratory syndrome coronavirus 2; ZIKV, Zika virus.

**Table 2 viruses-15-02333-t002:** Functions of EVs containing viral material with respect to five different pro-viral strategies.

Strategy	Proviral Mechanism	Viral Components Contained in EVs	Reference
Trojan horse	Immune evasion	Virion of HAV	[31]
		virion of SARS-CoV-2	[37]
		virion of HCV	[39]
		virion of JCPyV	[34]
	Increased MOI	virions of Rotavirus and Norovirus	[103]
		virions of BKPyV	[36]
		virions of PV	[33]
	Non-lytic egress for non-enveloped viruses	Virions of CVB3	[32]
Alternative mode of infection	EV-mediated transfer of viral genome	gDNA of oncolytic AdV	[59]
		gRNA of DENV2	[22]
		gRNA of HCV	[55]
		gRNA of HPgV	[56]
Tropism expansion	EV-mediated cell entry	JCPyV virion	[35]
	EV-mediated diffusion	EV-mediated translocation through blood-brain barrier	[104]
Cell physiology alteration	Hamper immunity	Nef protein of HIV-1	[46,105]
		VP40 protein of EBOV	[44]
		DENV subgenomic flaviviral RNA	[66]
	Induce cell permissiveness	Nef protein of HIV-1	[47]
		miRNA and RNA TAR element of HIV-1	[63,64]
	Evasion from humoral immunity	Nef protein of HIV-1	[106]
		miRNA of HSV-1	[62]
	Activation of EV biogenesis	Nef protein of HIV-1	[107]
	Aggravate pathology	Hbx protein and mRNA of HCV	[50]
		LMP-1 protein and mRNA of EBV	[49,61,108]
Humoral immunity decoy	Sequestration of neutralizing antibodies	Spike protein of SARS-CoV-2	[41]
		GP protein of EBOV	[43]
		E protein of ZIKV	[24]

Virus abbreviations: AdV, Adenovirus; BKPyV, BK Polyomavirus; CVB3, Coxsackievirus B3; DENV2, dengue 2 virus; EBOV, Ebola virus; EBV, Epstein Barr virus; HAV, Hepatitis A virus; HCV, Hepatitis C virus; HIV, Human Immunodeficiency virus; HPgV, Human pegivirus; HSV, Herpex simplex 1 virus; JCPyV, JC polyomavirus; PV, Poliovirus; SARS-CoV-2, severe acute respiratory syndrome coronavirus 2; ZIKV, Zika virus.

## Data Availability

Not applicable.

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
