# Peer review of "Viral Components Trafficking with(in) Extracellular Vesicles"

_viruses, 2023, doi:10.3390/v15122333_

Round 1

Reviewer 1 Report (Previous Reviewer 2)

Comments and Suggestions for Authors

thank the authors for incorporating our suggestions. looking forward to see this published.

Author Response

We thank the reviewer for acknowledging our efforts in answering his/her fruitful comments

Reviewer 2 Report (New Reviewer)

Comments and Suggestions for Authors

Overall, this is an interesting review on the role of EVs in viral infection.

Please find my suggestions in the attached pdf.

Comments on the Quality of English Language

Please find my suggestions in the attached pdf.

Author Response

We are extremely thankful for the dedication of this reviewer to improving the manuscipt. We have made all the requested changes. His/her effort in correcting the text was very much appreciated and greatly improved the manuscript.

Reviewer 3 Report (New Reviewer)

Comments and Suggestions for Authors

The authors are to be commended in summarizing the data on the interactions of extracellular vesicles (EVs) with viruses or viral components, and how EVs can help viruses to modify infection process. I have not seen similar summary in the literature on this topic. The review is quite well-structured, but I have a few remarks:

Major points:

-In the text of MS the authors use the term “proviral” in a very unconventional sense, which may confuse the reader. In virology this word is more often used to describe the interaction of the virus genome with the cell genome. I also did not find that in the publications cited in Table 2, the term “proviral” was used in the same sense that the authors put into this term in this review.

-It would be very good if the authors, when introducing commonly used virus names for the first time, indicated in parentheses the name of the virus according to the current ICTV release.

 Minor points:

-Figure 1, part of the figure with symbols and their explanation. Naked and enveloped viruses should not be indicated by the same symbol.

-The names of viruses’ families should be written in italics.

-All abbreviations used in the tables should be explained in captions.

-Table 1, the phrase “(Pelleted at 70,000g to 400,000g depending on papers and association with exosome markers)” should be clarified and rephrased. It would be better to use “in different studies” instead of “depending on papers”.

-Hepatitis A virus appears in the text under the names Hepatovirus A and Hepatitis A virus, it would be to give both names the very first time the virus is mentioned.

-In most cases, the authors write the names of viruses with a capital letter but also in the text there are names of the viruses with a small letter. Examples: line 291; Table 1, the name of encephalomyocarditis virus.

-There is a problem in the display of Table 1, line where reference 68 is cited.

Author Response

We thank the reviewer for his/her insights that improved the manuscript.

For major points:
-there is indeed a confusion with the term "proviral". Whereas proviral refers to proviruses which integrate virus genetic material into the genome of a host cell, in our manuscript we used proviral to refer to conditions that favor the virus multiplication. To clarify the term, we now write "pro-viral" and define its meaning in its first usage.

-with regards to virus abbreviations, following the suggestion of reviewer 2, we now provide abbreviations of virus names under the tables. We believe that this lightens the text while simplifying access to the abbreviation meanings.

Minor comments:

-we have added another symbol for naked viruses
-the names of viruses' family are not italicized
-We have detailed the abbreviations in each table in the caption.
-we modified Table 1 as suggested.
-For clarification, we replaced Hepatovirus A with Hepatitis A virus throughout the text, thereby using only one name for this virus.
-Capitalizing virus names is not always required. It depends whether it is a common name or not. We have screened the text for inconsistency at this leve.
-thank you for pointing to this error. We have corrected it

This manuscript is a resubmission of an earlier submission. The following is a list of the peer review reports and author responses from that submission.

Round 1

Reviewer 1 Report

Comments and Suggestions for Authors

The authors in the paper “Viral components within extracellular vesicles” summarized how EV-virus interactions can shape infection focusing on different interesting aspects to stimulate research in EV-virus interactions. 

I report my considerations below:

·      The authors in the paragraph 2 “Viral components in EVs” should add a table reporting the most important literature data relating to the points described, to make reading easier and more attractive.

·      The authors in paragraph 3 “Fate of viral components associated with EVs” should add a table reporting the most important literature data relating to the points described, to make reading easier and more attractive. 

·      In my opinion the authors have to enrich the content of the paragraph 3 “Fate of viral components associated with EVs”. For example regarding the paragraph 3.1 “Biofluid dissemination” the authors should report more data regarding the use of EVs as disease markers for diagnostic in the field of viral diseases. I believe that it could be an interesting point for readers. Moreover, a paragraph concerning the data reported in the literature regarding the EVs and response to therapy in the field of viral diseases could be interesting for the readers.

·      In my opinion, the state of the art regarding the importance of isolating characterized pure EVs, especially devoid of viruses, to study EV functions on viral infection could be a point of fundamental importance for experts in the sector. The current bibliography regarding the different techniques used could be of considerable interest for the readers.

·      The authors need to formulate this sentence better: As a result, the non- enveloped HAV is secreted as a quasi-enveloped virus with EV-like membranes (line 115-116).

·      The authors have to specify the meaning of “lEVs” in the sentence “The Spike protein of SARS-CoV-2, and the envelope protein of HIV-1 and Zika virus were detected in uncharacterized EVs, while the Ebola virus glycoprotein was found in lEVs related to ectosomes (line 125-126)”.

·      The authors have to check the space between It and is in the sentence “it is possible for the ZIKV Non-Structural 1 (NS1) protein to bind the EV surface” (line 135-136).

·      The authors in the line 236 have to check the space between that and enhance in the sentence “Together with viral genomes, EVs can also contain viral cofactors (i.e., proteins) that enhance EV infectivity”. 

·      The authors have to formulate in a different way this sentence: “EVs can deliver diverse viral components in bystander cells or distant tissues to reconfigure cellular environment, most  of the time in favor of upcoming viral infection”. 

·      In the line 267 the authors have to correct the space between LMP-1 and [50,66]. 

Comments on the Quality of English Language

A Moderate editing of English language is required

Reviewer 2 Report

Comments and Suggestions for Authors

Here Felix and colleagues give a comprehensive overview of the various ways that EV interact with viruses, including during biogenesis, dissemination, uptake, and replication. 

To sketch the complete picture, the authors should cover a few examples how EV and viral biogenesis interact. For example, how HIV gag hijacks EV biogenesis by binding TSG101. Consequently, the authors could edit their current title to "trafficking with(in) EVs. 

Last minor comment is that the authors should stress that although EVs are detected in different biofluids that this not necessary mean that EVs can freely circulate through the body. Most EVs that are injected in the bloodstream end up in liver and kidney followed by some spread to lung and spleen (M. Kang et al JEV 2021).